# DISTRIBUTED METHODS WITH COMPRESSED COMMUNICATION FOR SOLVING VARIATIONAL INEQUALITIES, WITH THEORETICAL GUARANTEES

## ABSTRACT

Variational inequalities in general and saddle point problems in particular are increasingly relevant in machine learning applications, including adversarial learning, GANs, transport and robust optimization. With increasing data and problem sizes necessary to train high performing models across various applications, we need to rely on parallel and distributed computing. However, in distributed training, communication among the compute nodes is a key bottleneck during training, and this problem is exacerbated for high dimensional and over-parameterized models. Due to these considerations, it is important to equip existing methods with strategies that would allow to reduce the volume of transmitted information during training while obtaining a model of comparable quality. In this paper, we present the first theoretically grounded distributed methods for solving variational inequalities and saddle point problems using compressed communication: MASHA1 and MASHA2. Our theory and methods allow for the use of both unbiased (such as Rand$k$; MASHA1) and contractive (such as Top$k$; MASHA2) compressors. We empirically validate our conclusions using two experimental setups: a standard bilinear min-max problem, and large-scale distributed adversarial training of transformers.

## 1 INTRODUCTION

### 1.1 THE EXPRESSIVE POWER OF VARIATIONAL INEQUALITIES

Due to their abstract mathematical nature and the associated flexibility they offer in modeling various practical problems of interests, *variational inequalities (VI)* have been an active area of research in applied mathematics for more than half a century (Korpelevich, 1976; Harker & Pang., 1990; Facchinei & Pang, 2003). It is well known that VIs can be used to formulate and study convex optimization problems, convex-concave saddle point problems and games, for example, in an elegant unifying mathematical framework (Korpelevich, 1976; Bauschke & Combettes, 2017).

Recently, Gidel et al. (2019) pointed out that multi-player games can be cast as VIs, and proposed to study mini-max or non-zero-sum games formulations of GANs (Goodfellow et al., 2014) in this fashion. This allowed them to successfully transfer established insights and well-known techniques from the vast literature on VIs to the study of GANs. In particular, oscillatory behavior of optimization methods (such as SGD) not originally designed to solve VI problems is well understood in the VI literature, and established tools, such as averaging and extrapolation, can be successfully applied to the training of GANs. Besides their usefulness in studying GANs and alternative adversarial learning models (Madry et al., 2018), VIs have recently attracted considerable attention of the machine learning community due to their ability to model other situations where the minimization of a single loss function does not suffice, such as auction theory (Syrgkanis et al., 2015) and robust and multi-agent reinforcement learning (Pinto et al., 2017).

In summary, VIs have recently become a potent tool enabling new advances in practical machine learning situations reaching beyond supervised learning where optimization problems and techniques, which can be seen as special instances of VIs and methods for solving them, reign supreme.

## 1.2 TRAINING OF SUPERVISED MODELS VIA DISTRIBUTED OPTIMIZATION

On the other hand, in the domain of classical, and hence also much better understood, *supervised machine learning* characterized by the fact that standard optimization techniques apply and work well, researchers and practitioners face other challenges that are currently beyond the reach of existing VI methods. Indeed, the training of modern supervised machine learning models in general, and deep neural networks in particular, is still extremely challenging. Due to their desire to improve the generalization of deployed models, machine learning engineers need to rely on training datasets of ever increasing sizes and on elaborate over-parametrized models (Arora et al., 2018). Supporting workloads of such unprecedented magnitudes would be impossible without combining the latest advances in hardware acceleration, distributed systems and *distributed algorithm design* (Verbraeken et al., 2019).

When training such modern supervised models in a distributed fashion, *communication cost* is often the bottleneck of the training system, and for this reason, a lot of effort was recently targeted at the design of communication efficient distributed optimization methods (Konečný et al., 2016; Smith et al., 2018; Ghosh et al., 2020; Gorbunov et al., 2021). A particularly successful technique for improving the communication efficiency of distributed first order optimization methods is *communication compression*. The idea behind this technique is rooted in the observation that in practical implementations it often advantageous to communicate messages compressed via (often randomized) *lossy compression techniques* instead of communicating the full messages (Seide et al., 2014; Alistarh et al., 2017). If the number of parallel workers is large enough, the noise introduced by compression is reduced, and training with compressed communication will often lead to the comparable test error while reducing the amount of communicated bits, which results in faster training, both in theory and practice (Mishchenko et al., 2019; Gorbunov et al., 2021).

## 1.3 TWO CLASSES OF COMPRESSION OPERATORS

We say that a (possibly) stochastic mapping $Q : \mathbb{R}^d \to \mathbb{R}^d$ is an *unbiased compression operator* if there exists a constant $q \geq 1$ such that

$$\mathbb{E}Q(z) = z, \quad \mathbb{E}\|Q(z)\|^2 \leq q\|z\|^2, \quad \forall z \in \mathbb{R}^d. \tag{1}$$

Further, we say that a stochastic mapping $C : \mathbb{R}^d \to \mathbb{R}^d$ is a *contractive compression operator* if there exists a constant $\delta \geq 1$ such that

$$\mathbb{E}\|C(z) - z\|^2 \leq (1 - 1/\delta)\|z\|^2, \quad \forall z \in \mathbb{R}^d. \tag{2}$$

If $b$ is the number of bits needed to represent a single float (e.g., $b = 32$ or $b = 64$), then the number of bits needed to represent a generic vector $z \in \mathbb{R}^d$ is $\|z\|_{\text{bits}} := bd$. To describe how much a compression operator reduces its input vector on average, we define the notion of expected density, defined via $\beta := \frac{1}{bd}\mathbb{E}\|Q(z)\|_{\text{bits}}$, where $\|Q(z)\|_{\text{bits}}$ denotes the number of bits needed to represent the quantized vector $Q(z)$. Note that $\beta \leq 1$. For the Rand$k$ operator (Alistarh et al., 2018; Beznosikov et al., 2020) we have $q = d/k$ and $\beta = k/d$.

## 1.4 TOWARDS COMMUNICATION-EFFICIENT DISTRIBUTED METHODS FOR VIS

While classical VI algorithms, such as the *extragradient method* originally proposed by Korpelevich (1976) and later studied by many authors (Nemirovski, 2004; Juditsky et al., 2008), were *not* designed to work in a distributed environment, virtually all methods that were (Yuan et al., 2014; Hou et al., 2021; Deng & Mahdavi, 2021; Beznosikov et al., 2021b;c) do *not* consider the general VI problem, but tackle the special case of saddle point problems only. Moreover, none of these distributed methods support communication compression, with the exception of the work of Yuan et al. (2014), which relies on rounding to the nearest integer multiple of a certain quantity. This compression mechanism does not offer theoretical benefits and does not even lead to convergence to the solution due to the errors introduced through rounding persist and prevent the method from solving the problem.

## 2 SUMMARY OF CONTRIBUTIONS

In this paper, we investigate whether it is possible to design communication-efficient algorithms for solving distributed VI problems by borrowing generic communication compression techniques

Table 1: An overview of existing methods and their high-level properties. We develop the first provably communication-efficient (via communication compression) algorithms for solving distributed VI problems.

| Reference | Solves general VIs? | Supports distributed setup? | Supports Compressed Communication? | Has strong theory? |
|---|---|---|---|---|
| Korpelevich (1976) | ✓ | ✗ | ✗ | ✓ |
| Nemirovski (2004) | ✓ | ✗ | ✗ | ✓ |
| Goodfellow et al. (2014) | ✗[8] | ✗ | ✗ | ✗[9] |
| Yuan et al. (2014) | ✗[1] | ✓[6] | ✗[7] | ✗[4] |
| Hou et al. (2021) | ✗[1] | ✓ | ✗[3] | ✗[5] |
| Deng & Mahdavi (2021) | ✗[1] | ✓ | ✗[3] | ✗[5] |
| Beznosikov et al. (2021b) | ✗[1] | ✓ | ✗[2] | ✓ |
| Beznosikov et al. (2021c) | ✗[1] | ✓ | ✗[2] | ✓ |
| Beznosikov et al. (2021a) | ✓ | ✓[6] | ✗[2] | ✓ |
| **This work** | ✓ | ✓ | ✓ | ✓ |

[1] Studies Saddle Point Problems (SPPs) only.
[2] Achieves communication efficiency without compression but by assuming *data similarity/homogeneity*; i.e., by dramatically restricting the problem class.
[3] Tries to achieve communication efficiency via performing local steps as popular in Federated Learning (Konečný et al., 2016; McMahan et al., 2016; Li et al., 2020). Employs local methods (such as GD and SCAFFOLD (Karimireddy et al., 2020)).
[4] Does not achieve convergence to the solution; and the convergence criterion may not work for simple bilinear problems.
[5] They build method on GD, which diverges for simple bilinear problems.
[6] Uses decentralized architecture.
[7] Compression just by rounding to nearest integer multiples of some constant.
[8] Applies to GANs only.
[9] No theory.

(1) and (2) from the optimization literature (Seide et al., 2014; Alistarh et al., 2017; Mishchenko et al., 2019; Gorbunov et al., 2021; Richtárik et al., 2021) and adapting and embedding them into established and efficient methods for solving VIs (Korpelevich, 1976; Nemirovski, 2004; Juditsky et al., 2008; Alacaoglu & Malitsky, 2021). Whether or not this is possible is an open problem.

In summary, we design the first provably communication-efficient algorithms for solving general distributed VI problems (see Section 3, Equation 3) in the deterministic (see (4)) and stochastic (see (5)) regimes, supporting both unbiased (MASHA1 = Algorithm 1) and contractive (MASHA2 = Algorithm 2) compressors. Our methods are explicitly designed to be variance reduced to achieve better theoretical properties and better practical performance.

In Table 1 we give a high level overview of existing methods for VIs, and contrast them with our methods and results. We now elaborate a bit more:

## 2.1 TWO DISTRIBUTED PROBLEMS: DETERMINISTIC AND STOCHASTIC

We study two distributed VI problems: i) *deterministic*, where the monotone operator $F : \mathbb{R}^d \to \mathbb{R}^d$ featured in the VI is the average of $M$ operators $\{F_m\}_{m=1}^M$, where $M$ is the number of devices/machines, which can be evaluated in each communication round, and ii) *stochastic*, where each monotone operator $F_m : \mathbb{R}^d \to \mathbb{R}^d$ has a finite-sum structure on its own, and only a single operator in the sum can be evaluated in each iteration. In contrast to previous works, we study general constrained VIs in the distributed setup (see Section 3), and not merely saddle point problems.

## 2.2 TWO NEW METHODS WITH COMPRESSED COMMUNICATION: MASHA1 AND MASHA2

We develop two extensions of the *extragradient / extrastep method* of Korpelevich (1976) to distributed VIs depending on whether we use unbiased (1) or contractive (2) compressors, since each type of compressor demands a different algorithmic design and a different analysis. In particular, contractive compressors are notoriously hard to analyze even for optimization problems (Karimireddy et al., 2019; Richtárik et al., 2021). Our method based on unbiased compressors is called MASHA1 (Algorithm 1), and our method based on contraction compressors is called MASHA2 (Algorithm 2). Both are designed to handle the deterministic and also the stochastic setting, and both are enhanced with bespoke *variance-reduction* techniques for better theoretical and practical performance. Due to space

Table 2: Summary of our **iteration complexity** results for finding an $\varepsilon$-solution for problem (3) in the deterministic (i.e., (4)) and stochastic (i.e., (4)–(5)) setups. In the strongly convex - strongly convex case, convergence is measured by the distance to the solution. In the convex-concave case, convergence is measured in terms of the gap function. *Notation:* $\mu$ = sum of the coefficients $\mu_F$ and $\mu_h$ of strong monotonicity of the operator $F$ and strong convexity of $h$, $L$ = maximum of local Lipschitz constants $L_m$, $R$ = diameter (in Euclidean norm) of the optimization set, $R_0$ = initial distance to the solution, $q$ = the variance parameter associated with an unbiased compressor (see (1)); $\delta$ = the variance parameter associated with a contractive compressor (see (2)); $M$ = the number of parallel clients/nodes; $r$ = the size of the local dataset (see (5)). To simplify the bounds, we assume that compression occurs only on the nodes, and the server transmits full information, additionally, we assume that the expected density $\beta$ of the compression operators satisfies $q = 1/\beta$ and $\delta = 1/\beta$ (e.g., this holds for Rand$k$ and Top$k$).

| Problem | Algorithm | Strongly convex - strongly concave case |
|---------|-----------|------------------------------------------|
| (3)–(4) | MASHA1 | $\left[1 + q + \frac{L}{\mu}\sqrt{1 + q + \frac{q^2}{M}}\right] \cdot \log\frac{R_0^2}{\varepsilon}$ |
|         | MASHA2 | $\left[1 + \delta + \frac{\delta^{3/2}L}{\mu} + \frac{\delta^3 L^2}{\mu^2}\right] \cdot \log\frac{R_0^2}{\varepsilon}$ |
| (3)–(5) | MASHA1 | $\left[\max(r, q+1) + \frac{L}{\mu}\sqrt{\max(r, q+1)\left(1 + \frac{q}{M}\right)}\right] \cdot \log\frac{R_0^2}{\varepsilon}$ |
|         | MASHA2 | $\left[\max(r, \delta+1) + \sqrt{\max(r, q+1)}\frac{\delta L}{\mu} + \max(r, q+1)\frac{\delta^2 L^2}{\mu^2}\right] \cdot \log\frac{R_0^2}{\varepsilon}$ |
| **Problem** | **Algorithm** | **Convex-concave case** |
| (3)–(4) | MASHA1 | $\sqrt{1 + q + \frac{q^2}{M}} \cdot \frac{LR^2}{\varepsilon}$ |
|         | MASHA2 | $-$ |
| (3)–(5) | MASHA1 | $\max(r, \delta+1)\sqrt{1 + \frac{q}{M}} \cdot \frac{LR^2}{\varepsilon}$ |
|         | MASHA2 | $-$ |

restrictions, we only describe MASHA1 in the main body of the paper, and relegate MASHA2 and the associated theory to Appendix B.

## 2.3 THEORETICAL COMPLEXITY RESULTS

We establish a number of theoretical complexity results for our methods, which we summarize in Table 2. We consider the strongly convex - strongly concave regime as well as the more general convex - concave regime. In the first case we obtain linear convergence results ($O(\log 1/\epsilon)$) in terms of the distance to solution, and in the latter we obtain fast sublinear convergence results ($O(1/\epsilon)$) in terms of the *gap* function. To get an estimate for the number of information transmitted, one need to multiply the estimates from Table 1 by $1/q$ and $1/\delta$, respectively. Then we get that from the point of view of the transmitted information, MASHA1 is better by a factor $\sqrt{1/q + 1/M}$ in comparison with the original extragradient.

## 3 PROBLEM FORMULATION AND ASSUMPTIONS

Let us first introduce basic notation. We write $\langle x, y \rangle := \sum_{i=1}^d x_i y_i$ to denote the standard Euclidean inner product of vectors $x, y \in \mathbb{R}^d$. This induces $\ell_2$-norm in $\mathbb{R}^d$ as usual: $\|x\| := \sqrt{\langle x, x \rangle}$. We also introduce the proximal operator, defined as $\text{prox}_g(z) := \arg\min_{u \in \mathcal{Z}}\{g(u) + \frac{1}{2}\|u - z\|^2\}$, which is well defined for proper lower semicontinuous convex functions $g : \mathbb{R}^d \to \mathbb{R} \cup \{+\infty\}$.

### 3.1 PROBLEM FORMULATION

We study distributed variational inequality (VI) problems of the form

$$\text{Find} \quad z^* \in \mathcal{Z} \quad \text{such that} \quad \langle F(z^*), z - z^* \rangle + h(z) - h(z^*) \geq 0, \quad \forall z \in \mathcal{Z}, \tag{3}$$

where $\mathcal{Z}$ is a nonempty closed convex subset of $\mathbb{R}^d$, $F : \mathbb{R}^d \to \mathbb{R}^d$ is an operator with certain favorable properties (e.g., Lipschitzness and monotonicity), and $h : \mathbb{R}^d \to \mathbb{R} \cup \{+\infty\}$ is a proper lower semicontinuous convex function. We assume that the training data describing $F$ is *distributed*

across $M$ workers/nodes/clients

$$F(z) := \frac{1}{M} \sum_{m=1}^{M} F_m(z), \tag{4}$$

where $F_m : \mathbb{R}^d \to \mathbb{R}^d$ for all $m \in \{1, 2, \dots, M\}$, and for some results we further assume that $F_m$ is of a finite-sum structure as well:

$$F_m(z) := \frac{1}{r} \sum_{i=1}^{r} F_{m,i}(z). \tag{5}$$

## 3.2 Assumptions

Next, we list two key assumptions - both are standard in the literature on VIs.

**Assumption 1 (Lipschitzness in mean)** *For all clients $m = 1, 2, \dots, M$, the operator $F_m(z) : \mathbb{R}^d \to \mathbb{R}^d$ is Lipschitz in mean with constant $L_m \geq 0$ on $\mathcal{Z}$. That is,*

$$\frac{1}{r} \sum_{i=1}^{r} \|F_{m,i}(z_1) - F_{m,i}(z_2)\|^2 \leq L_m^2 \|z_1 - z_2\|^2, \quad \forall z_1, z_2 \in \mathcal{Z}. \tag{6}$$

For problem (4), Assumption 1 is to be interpreted to hold with $r$ is equal 1, i.e., with $F_m = F_{m,1}$.

**Assumption 2 (Monotonicity and convexity)** *(SM) Strong monotonicity/strong convexity. There exist non-negative constants $\mu_F, \mu_h$ such that $\mu_h + \mu_F > 0$, and the following statements hold:*

$$\langle F(z_1) - F(z_2), z_1 - z_2 \rangle \geq \mu_F \|z_1 - z_2\|^2, \quad \forall z_1, z_2 \in \mathcal{Z}, \tag{7}$$

$$h(z_1) - h(z_2) - \langle \nabla h(z_2), z_1 - z_2 \rangle \geq \frac{\mu_h}{2} \|z_1 - z_2\|^2, \quad \forall z_1, z_2 \in \mathcal{Z}. \tag{8}$$

*(M) Monotonicity/convexity.*

$$\langle F(z_1) - F(z_2), z_1 - z_2 \rangle \geq 0, \quad h(z_1) - h(z_2) - \langle \nabla h(z_2), z_1 - z_2 \rangle \geq 0, \quad \forall z_1, z_2 \in \mathcal{Z}. \tag{9}$$

## 4 MASHA1: Handling Unbiased Compressors

In this section we present only one of our two new algorithms: MASHA1 (Algorithm 1) - the method that relies on unbiased compressors. Due to lack of space, we include MASHA2 (Algorithm 2) - the method that relies on contractive compressors in the appendix. Both algorithms are presented for two modes: deterministic (4) and stochastic (4)–(5). We denote the lines related to the deterministic regime in blue, and in red - to the stochastic one. Black lines are common to both modes.

### 4.1 The Algorithm

MASHA1 is a modification of the extrastep method. At the beginning of each iteration of Algorithm 1, each device knows the value of $F(w^k)$, hence it can be calculated $\bar{z}^k$ locally. $\bar{z}^k$ is a convex combination of $z^k$ and $w^k$ with momentum parameter $\tau$ (typically, $\tau$ is close to 1). Also Algorithm can compute $z^{k+1/2}$ locally. Then it sends the compressed version of the difference $F_m(z^{k+1/2}) - F_m(w^k)$ to the server, and the server does a reverse compressed broadcast. As a result, an unbiased estimate of $F(z^{k+1/2}) - F(w^k)$ appears on each node. Also, the nodes receive a bit of information $b_k$. This bit is generated randomly on the server and is equal to 0 with probability $\tau$. Then, all devices make a final update on $z^{k+1}$, and also update the $w^{k+1} = z^{k+1}$ point if the $b_k = 1$ or save it from the previous iteration $w^{k+1} = w^k$. In the case when the point $w^{k+1} = z^{k+1}$, we need to exchange the full values of $F_m(w^{k+1})$ in order that at the beginning of the next iteration the value of $F(w^{k+1})$ is known to all nodes. In the stochastic case, Algorithm 1 has the same form, the only thing that changes is that one need to generate a function number (batch number) from 1 to $r$. We use a possibly difference compressor on each device and also on the server. To distinguish between them, we will use the following notation: $Q_m^{\text{dev}}, q_m^{\text{dev}}, \beta_m^{\text{dev}}$ and $Q^{\text{serv}}, q^{\text{serv}}, q^{\text{serv}}$. Note that, if $Q$ is the identity quantization, i.e., $Q(x) = x$, then MASHA1 is a distributed analogue of the method from Alacaoglu & Malitsky (2021).

It is important to note main differences from minimization problems. For minimization problems, compressed gradient methods are constructed on the gradient descent or the accelerated gradient

---

**Algorithm 1** MASHA1 (handling unbiased compressors)

---

**Parameters:** Stepsize $\gamma > 0$, number of iterations $K$.
**Initialization:** Choose $z^0 = w^0 \in \mathcal{Z}$.
Server sends to devices $z^0 = w^0$ and devices compute $F_m(w^0)$ and send to server and get $F(w^0)$
**for** $k = 0, 1, 2, \ldots, K-1$ **do**
    **for** each device $m$ in parallel **do**
        $\bar{z}^k = \tau z^k + (1 - \tau) w^k$
        $z^{k+1/2} = \text{prox}_{\gamma h}(\bar{z}_k - \gamma F(w^k))$,
        Generate $\pi_m^k$ from $\{1, \ldots, r\}$ independently
        Compute $F_m(z^{k+1/2})$ & send $Q_m^{\text{dev}}(F_m(z^{k+1/2}) - F_m(w^k))$ to server
        Compute $F_{m,\pi_m^k}(z^{k+1/2})$ & send $Q_m^{\text{dev}}(F_{m,\pi_m^k}(z^{k+1/2}) - F_m(w^k))$ to server
    **end for**
    **for** server **do**
        Compute $Q^{\text{serv}}\left[\frac{1}{M}\sum\limits_{m=1}^{M} Q_m^{\text{dev}}(F_m(z^{k+1/2}) - F_m(w^k))\right]$ & send to devices
        Compute $Q^{\text{serv}}\left[\frac{1}{M}\sum\limits_{m=1}^{M} Q_m^{\text{dev}}(F_{m,\pi_m^k}(z^{k+1/2}) - F_{m,\pi_m^k}(w^k))\right]$ & send to devices
        Sends to devices one bit $b_k$: 1 with probability $1 - \tau$, 0 with with probability $\tau$
    **end for**
    **for** all devices in parallel **do**
        $z^{k+1} = \text{prox}_{\gamma h}\left(\bar{z}_k - \gamma Q^{\text{serv}}\left[\frac{1}{M}\sum\limits_{m=1}^{M} Q_m^{\text{dev}}(F_m(z^{k+1/2}) - F_m(w^k))\right] - \gamma F(w^k)\right)$
        $z^{k+1} = \text{prox}_{\gamma h}\left(\bar{z}_k - \gamma Q^{\text{serv}}\left[\frac{1}{M}\sum\limits_{m=1}^{M} Q_m^{\text{dev}}(F_{m,\pi_m^k}(z^{k+1/2}) - F_{m,\pi_k^m}(w^k))\right] - \gamma F(w^k)\right)$
        **if** $b_k = 1$ **then**
            $w^{k+1} = z^{k+1}$
            Compute $F_m(w^{k+1})$ & send it to server; and get $F(w^{k+1})$ as a response from server
        **else**
            $w^{k+1} = w^k$
        **end if**
    **end for**
**end for**

---

descent methods. Here, the extragradient method is taken as a basis. In the experiments, we will see the importance of this fact, i.e., gradient descent type methods will diverge even on simple problems. The second key difference is that we need not compress $F_m(z^{k+1/2})$ itself, but the difference $F_m(z^{k+1/2}) - F(w^k)$. We will also see the importance of this approach in the experiments. The later idea is similar to the approach used in the DIANA, VR-DIANA (Mishchenko et al., 2019; Horváth et al., 2019), MARINA (Gorbunov et al., 2021) and EF21 (Richtárik et al., 2021) methods in optimization.

## 4.2 THEORY

We now establish convergence of MASHA1 in both regimes: deterministic and stochastic (finite-sum). Our analysis relies on the following *Lyapunov function*:

$$V_k := \tau\|z^k - z^*\|^2 + \|w^k - z^*\|^2, \tag{10}$$

This criterion is used in the strongly monotone case. For the general monotone case, another convergence criterion is used - the *gap function*:

$$\text{Gap}(z) := \sup_{u \in \mathcal{C}} \left[\langle F(u), z - u \rangle + h(z) - h(u)\right]. \tag{11}$$

Here we do not take the maximum over the entire set $\mathcal{Z}$ (as in the classical version), but over $\mathcal{C}$ – a compact subset of $\mathcal{Z}$. Thus, we can also consider unbounded sets $\mathcal{Z}$. This is permissible, since such a version of the criterion is valid if the solution $z^*$ lies in $\mathcal{C}$; for details see the work of Nesterov (2007).

**Theorem 1 (Convergence of MASHA1)** *Let Assumption 1 be satisfied. Then, if one of cases from Assumption 2 is additionally fulfilled, the following estimates hold for the iterates of* MASHA1:

- *for strongly-monotone/convex case with stepsize $0 < \gamma \leq \min\left\{\frac{\sqrt{1-\tau}}{4C_q}; \frac{1-\tau}{4(\mu_F+\mu_h)}\right\}$ and $C_q :=$*
$\sqrt{\frac{q^{serv}}{M^2}\sum_{m=1}^{M}(q_m^{dev} + M - 1)L_m^2}$:

$$\mathbb{E}\left[V_K\right] \leq \left(1 - \gamma \cdot \frac{\mu_F + \mu_h}{16}\right)^{K-1} \cdot V_0,$$

- *for monotone/convex case with $\gamma \leq \frac{\sqrt{1-\tau}}{4C_q}$:*

$$\mathbb{E}\left[Gap(\bar{z}^K)\right] \leq \frac{8\max_{u\in\mathcal{C}}\left[\|z^0-u\|^2\right]}{\gamma K}, \quad where \quad \bar{z}^K = \frac{1}{K}\sum_{k=0}^{K-1} z^{k+1/2}.$$

For proof, see Appendix A.

An important issue of convergence is the choice of $\tau$. If $\tau = 0$ we have the fastest convergence rate, on the other hand, this means that we must send full $F_m$ in each iteration (because we update $w^k$). This is disadvantageous from a communication point of view. The next corollaries give the rules for the right choice of $\tau$, as well as the iterative (in deterministic case), oracle (in stochastic) and communication complexities of Algorithm 1 in both cases. The method without quantization (for example, the ordinary distributed extragradient method) in one iteration transmits the number of information proportional to $\Omega(bdM)$ bits, then here we measure the communication complexity in terms of $\Omega(bdM)$. In literature about method with compression, it is typical to consider only devices compression (Alistarh et al. (2017); Mishchenko et al. (2019); Horváth et al. (2019); Gorbunov et al. (2021); Beznosikov et al. (2020)), because it is easier for server to broadcast information. In the next corollary we consider the transfer of information in one direction only (from devices to the server). It means we choose $q^{serv} = 1$ and $\beta^{serv} = 1$.

**Corollary 1 (Convergence of MASHA1 in the deterministic case)** *Let the deterministic problem (3)–(4) be solved by MASHA1 with precision $\varepsilon$. Let Assumption 1, and one of the two cases from Assumption 2 be satisfied. If we choose*

$$\tau = 1 - \beta := 1 - \frac{1}{M}\sum_{m=1}^{M}\beta_m^{dev}, \tag{12}$$

*then we have the following estimates for the total # of iterations and the total # of transferred bits from devices to the server:*

- *in the strongly-monotone/convex case:*

$$\mathcal{O}\left(\left[\frac{1}{\beta} + \frac{C_q}{\sqrt{\beta}(\mu_F+\mu_h)}\right]\log\frac{2\|z^0-z^*\|^2}{\varepsilon}\right) \text{ iters}, \qquad \mathcal{O}\left(\left[1 + \frac{\sqrt{\beta}C_q}{\mu_F+\mu_h}\right]\log\frac{2\|z^0-z^*\|^2}{\varepsilon}\right) \text{ bits},$$

- *in the monotone/convex case*

$$\mathcal{O}\left(\frac{C_q\max_{u\in\mathcal{C}}\left[\|z^0-u\|^2\right]}{\sqrt{\beta}\varepsilon}\right) \text{ iters}, \qquad \mathcal{O}\left(\frac{\sqrt{\beta}C_q\max_{u\in\mathcal{C}}\left[\|z^0-u\|^2\right]}{\varepsilon}\right) \text{ bits}.$$

One can see that our Algorithm 1 can outperform the uncompressed extragradient method. Let us compare them in the monotone case. The communication complexity of the extragradient method is $\mathcal{O}\left(L\max_{u\in\mathcal{C}}\left[\|z^0-u\|^2\right]\varepsilon^{-1}\right)$. Let us consider the case when $L_m = L$ for all $m \in \{1, 2, \ldots, M\}$ and $q_m^{dev} = q^{dev}$. Then MASHA1 has communication complexity $\mathcal{O}\left(L\max_{u\in\mathcal{C}}\left[\|z^0-u\|^2\right]\varepsilon^{-1}\sqrt{\beta(1+q^{dev}/M)}\right)$. If $\beta(1+q^{dev}/M) < 1$, we outperform the standard uncompressed extragradient method. For example, for if we consider RandK operator $\beta = 1/q^{dev}$. This means that we have $\mathcal{O}\left(L\max_{u\in\mathcal{C}}\left[\|z^0-u\|^2\right]\varepsilon^{-1}\cdot\sqrt{1/q^{dev}+1/M}\right)$.

**Corollary 2 (Convergence of MASHA1 in the stochastic case)** *Let the stochastic problem (3)–(5) be solved by MASHA1 with precision $\varepsilon$. Let Assumption 1 and one of the two cases from Assumption 2 be satisfied. If we choose*

$$\tau = 1 - \min\left(\frac{1}{r}; \beta\right)$$

*then we have the following estimates for the total # of oracle calls and the total # of transferred bits:*

- *in the strongly-monotone/convex case:*

$$\mathcal{O}\left(\left[\max\left(r;\tfrac{1}{\beta}\right)+\max\left(\sqrt{r};\tfrac{1}{\sqrt{\beta}}\right)\tfrac{C_q}{(\mu_F+\mu_h)}\right]\log\tfrac{2\|z^0-z^*\|^2}{\varepsilon}\right)\ calls,$$

$$\mathcal{O}\left(\left[\max\left(1;r\beta\right)+\max\left(\sqrt{r\beta^2};\sqrt{\beta}\right)\tfrac{C_q}{\mu_F+\mu_h}\right]\log\tfrac{2\|z^0-z^*\|^2}{\varepsilon}\right)\ bits,$$

- *in the monotone/convex case*

$$\mathcal{O}\left(\max\left(r;\tfrac{1}{\beta}\right)\tfrac{C_q\max_{u\in\mathcal{C}}\left[\|z^0-u\|^2\right]}{\varepsilon}\right)\ calls,\quad \mathcal{O}\left(\max\left(\sqrt{r\beta^2};\sqrt{\beta}\right)\tfrac{C_q\max_{u\in\mathcal{C}}\left[\|z^0-u\|^2\right]}{\varepsilon}\right)\ bits.$$

In the case of a finite sum, in addition to the number of transmitted information, it is not iterations are important to us, but the number of calls to the oracle for $F_{m,r}$. This is due to the fact that it is $r$ times more expensive to calculate full $F_m$, and the calculation of $F_m$ should be avoided. For deterministic case (when $r = 1$) estimates from Corollary 2 is the same as in Corollary 1.

## 5 EXPERIMENTS

### 5.1 BILINEAR SADDLE POINT PROBLEM

We start our experiments with a distributed bilinear problem:

$$\min_{x\in\mathbb{R}^d}\max_{y\in\mathbb{R}^d}f(x,y):=\tfrac{1}{M}\sum_{m=1}^{M}f_m(x,y),\quad f_m(x,y):=x^\top A_m y+a_m^\top x+b_m^\top y+\tfrac{\lambda}{2}\|x\|^2-\tfrac{\lambda}{2}\|y\|^2,$$
(13)

where $A_m \in \mathbb{R}^{d\times d}$, $a_m, b_m \in \mathbb{R}^d$. This is a saddle point problem, and $F$ is written as follows: $F(x,y) := (\nabla_x f(x,y); -\nabla_y f(x,y))$. This operator is $\lambda$ strongly monotone and, moreover, all functions $F_m$ are $(\|A_m\|_2 + \lambda)$-Lipschitz. Therefore, such a distributed problem is well suited for the primary comparison of our methods. We take $d = 100$ and generate positive definite matrices $A_m$ and vectors $a_m, b_m$ randomly.

The purpose of the experiment is to understand whether the MASHA1 and MASHA2 methods are superior to those in the literature. As a comparison, we take QGD with natural dithering Horvath et al. (2019), classical error feedback with Top 30% compression, as well as an extra step method, each step of which is used with natural rounding. In MASHA1 (Algorithm 1) we also used natural dithering, in MASHA2 (Algorithm 2) – Top30%. See results in Figure 1.

Figure 1: Comparison MASHA1 (Algorithm 1) and MASHA2 (Algorithm 2) with existing methods in iterations and in Mbytes.

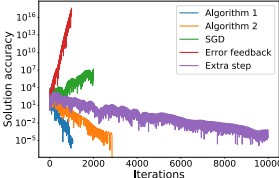 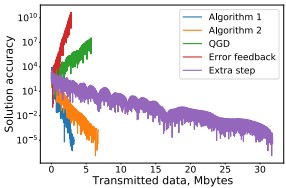

We see that methods based on gradient descent (QSG and EF) diverge. This confirms that one needs to use method specifically designed for saddle point problems (for example, the extragradient method), and not classical optimization methods. The much slower convergence of the quantized extragradient method shows the efficiency of our approach in which we compress the differences $F_m(z^{k+1/2}) - F(w^k)$.

### 5.2 ADVERSARIAL TRAINING OF TRANSFORMERS

We now evaluate how compression performs for variational inequalities (and for saddle point problems, as a special case) in a more practically motivated scenario. Indeed, saddle point problems (special case of variational inequalities) have sample applications in machine learning (see Appendix D), including *adversarial training*. We consider one of these tasks and train a *transformer-based masked*

*language model* (Vaswani et al., 2017; Devlin et al., 2019; Liu et al., 2019) using a fleet of 16 low-cost preemptible workers with T4 GPU and low-bandwidth interconnect. For this task, we use the compute-efficient adversarial training regimen proposed for transformers by Zhu et al. (2019); Liu et al. (2020). Formally, the adversarial formulation of the problem is the min-max problem

$$\min_{w} \max_{\|\rho_1\| \le e, \dots \|\rho_N\| \le e} \frac{1}{N} \sum_{n=1}^{N} l(f(w, x_n + \rho_n, y_n)^2 + \frac{\lambda}{2}\|w\|^2 - \frac{\beta}{2}\|\rho\|^2,$$

where $w$ are the weights of the model, $\{(x_n, y_n)\}_{n=1}^{N}$ are pairs of the training data, $\rho$ is the so-called adversarial noise which introduces a perturbation in the data, and $\lambda$ and $\beta$ are the regularization parameters. To make our setup more realistic, we train ALBERT-large with layer sharing (Lan et al., 2020), which was recently shown to be much more communication-efficient during training (Ryabinin et al., 2021; Diskin et al., 2021). We train our model on a combination of Bookcorpus and Wikipedia datasets with the same optimizer (LAMB) and parameters as in the original paper (Lan et al., 2020), use the adversarial training configuration of Zhu et al. (2019), and follow system design considerations for preemptible instances (Ryabinin et al., 2021).

In terms of communication, we consider 4 different setups for gradient compression: the "baseline" strategy with uncompressed gradients, full 8-bit quantization (Dettmers, 2015; Lin et al., 2018), mixed 8-bit quantization, and Power compression (Vogels et al., 2019) with rank $r=8$. For mixed 8-bit quantization and Power we only apply compression to gradient tensors with more than $2^{16}$ elements, sending smaller ones uncompressed. These small tensors represent layer biases and LayerNorm scales (Ba et al., 2016) that collectively amount to $\le 1\%$ of the total gradient, but can be more difficult to compress than regular weight tensors. Finally, since Power is a biased compression algorithm, we use error feedback (Karimireddy et al., 2019; Richtárik et al., 2021) with a modified formulation proposed by Vogels et al. (2019). For all experimental setups, we report learning curves in terms of the model training objective, similarly to (Fedus et al., 2021; Ryabinin et al., 2021). To better quantify the differences in training loss, we also evaluate the downstream performance for each model on several popular tasks from (Wang et al., 2018) after each model was trained on approximately 80 billion tokens. Finally, we measure the communication efficiency of each proposed strategy by measuring the average wall time per communication round when all 16 workers are active.

Figure 2: **(upper left)** ALBERT training objective convergence rate with different compression algorithms; **(upper right)** ALBERT training objective convergence rate with different compression algorithms (zoomed); **(lower)** Average wall time per communication round with standard deviation over 5 repetitions and downstream evaluation scores on GLUE benchmark tasks after at 80 billion training tokens ($\approx 10^4$ optimizer steps).

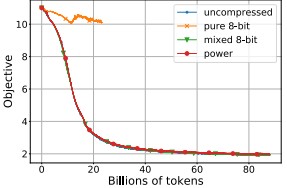 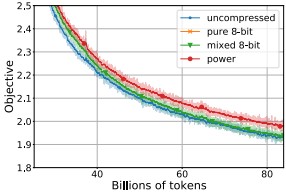

| Setup | Avg time | Avg score | CoLA | MNLI | MRPC | QNLI | QQP | RTE | SST2 | STS-B | WNLI |
|---|---|---|---|---|---|---|---|---|---|---|---|
| Baseline | $8.79 \pm 0.03$ | 71.96 | 45.2 | 81.1 | 83.0 | 88.3 | **89.0** | **67.8** | 85.5 | **89.4** | **18.3** |
| Full 8-bit | $4.42 \pm 0.07$ | N/A | N/A | N/A | N/A | N/A | N/A | N/A | N/A | N/A | N/A |
| Mixed 8-bit | $4.61 \pm 0.08$ | **72.12** | **48.8** | **81.3** | **88.7** | 88.1 | 85.2 | 64.3 | 88.3 | 87.5 | 16.9 |
| Power | $\mathbf{1.57 \pm 0.05}$ | 69.52 | 43.9 | 80.5 | 85.6 | **88.6** | 86.0 | 47.2 | **88.5** | 88.5 | 16.9 |

The learning curves in Figure 2 (upper) follow a predictable pattern, with more extreme compression techniques demonstrating slower per-iteration convergence. One curious exception to that is full 8-bit quantization, which was unable to achieve competitive training loss. The remaining three setups converge to similar loss values below 2. Both the baseline and mixed 8-bit compression show similar values in terms of downstream performance, with Power compression showing mild degradation. But in terms of information transfer time, methods using compression (especially Power) are significantly superior to the method without compression. This makes it possible to use such techniques to increase the training time without sacrificing quality.

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
