# OpenReview forum: "Distributed Methods with Compressed Communication for Solving Variational Inequalities, with Theoretical Guarantees"
_ICLR.cc/2022/Conference — ICLR 2022 Submitted_

### Official Review · Reviewer_oTML · 2021-11-01

**Correctness:** 2
**Technical Novelty And Significance:** 2
**Empirical Novelty And Significance:** 3
**Recommendation:** 3
**Confidence:** 4

**Main Review:**

The main weakness of this paper is regarding the theoretical results.

- The authors consider a general notion of unbiased compression schemes in (1). Parameter $q$ is in general dimension-dependent (Alistarh 2017). The authors establish the bounds by treating $q$ as a constant term. However, this parameter can be very large in overparameterized settings, which motivate this paper as stated in the abstract. When $q$ is large, $C_q$ in Theorem 1 will be large too, which leads to very small learning rate. Similar issue happens when $\tau$ is close to one. As authors mentioned $\tau$ should be close to one to control communication costs. When $\tau$ is close to one, the learning rate becomes very small, which does not lead to interesting results in terms of convergence. A similar issue happens in Corollaries 1 and 2.

- I think the discussion after Corollary 1 is not accurate. In particular, the condition under which Algorithm 1 outperforms the standard uncompressed extragradient method is restrictive. It is more likely that the standard uncompressed extragradient method outperforms Algorithm 1 in terms of communication complexity.

-  It will be nice if the authors provide concrete examples of interesting VI problems beyond saddle point problems in machine learning. It is also important to show that Assumptions 1 and 2 hold for such problems. Indeed, the experiments are also based on saddle point problems.

- It is unclear how all nodes have access to $F(w^{k+1})$ at the beginning of the iteration $k+1$ while only some nodes with $b_k=1$ compute $F_m(w^{k+1})$?

- In Table 1, the authors mentioned that ``(Deng & Mahdavi 2021) does not have strong theory since it cannot improve on non-compressed methods without assuming data homogeneity.'' I my view, a similar criticism applies to this work too.

- "we introduce the notion of expected density, defined ..." This is not new. A similar notion based on the expected number of required bits has been proposed in the literature e.g., QSGD, NUQSGD, signSGD ...

- The paper is not well-written. There are many typos that require a substantial revision. Some examples include: abstract, Section 1.4, footnotes in Table 1, first paragraph of Section 4.1, first paragraph of Section 5.2, ...

Minor comment:
- Section 4.2: "We now establish convergence of MASHA1 in both regimes": please specify "both regimes".

- Some terms such as $\tau$ and $b_k$ are defined only in Algorithm 1. It will be easier for readers if those terms are defined in the main text too.

**Summary Of The Paper:**

The authors have considered general distributed variational inequalities problems and proposed two methods using unbiased and contractive compressors.They have evaluated their methods on saddle point problems including large-scale adversarial training of transformers.



**Summary Of The Review:**

The paper has some interesting aspects for example large-scale adversarial training of transformers. However, given the issues regarding theoretical results, connection with related work, and clarify and presentation of the paper, I recommend rejection.

---

> ### Author Response · Authors · 2021-11-23
> **Question 1: small stepsizes**
>
> **Question 1:**
>
> > The authors consider a general notion of unbiased compression schemes in (1). Parameter $q$  is in general dimension-dependent (Alistarh 2017). The authors establish the bounds by treating $q$  as a constant term. However, this parameter can be very large in overparameterized settings, which motivate this paper as stated in the abstract. When $q$  is large, $C_q$  in Theorem 1 will be large too, which leads to very small learning rate. Similar issue happens when $\tau$  is close to one. As authors mentioned $\tau$ should be close to one to control communication costs. When $\tau$  is close to one, the learning rate becomes very small, which does not lead to interesting results in terms of convergence. A similar issue happens in Corollaries 1 and 2.
>
> **Response 1:**
>
> - **This is not an issue with our paper. It is clear that the reviewer is not familiar with the literature on quantized/compressed methods in optimization. In brief, compression introduces noise, which forces the stepsizes to be smaller, and this leads to an increase in the number of iterations/communications. This is true for every single optimization method on compressed communication (and we know many dozens of them) we know of (some examples below).**
>
> - The reason why compressed communication helps is because the **communication savings due to compression outweigh the loss which leads to an increase in the number of communication rounds. So, communication complexity improves on the non-compressed methods!**
>
> - Notice that it is not the stepsize that matters. Communication complexity matters. And our guarantees show that it is more efficient to compress than not. **See our commentary following Corollary 1.**
>
>
> Examples of papers on communication compression:
>
> - Dan Alistarh, Demjan Grubic, Jerry Li, Ryota Tomioka, and Milan Vojnovic. QSGD: Communication-efficient SGD via gradient quantization and encoding. -- NIPS 2017
>
> - Eduard Gorbunov, Dmitry Kovalev, Dmitry Makarenko, and Peter Richtárik. Linearly converging error compensated SGD. -- NIPS 2020
>
> - Eduard Gorbunov, Konstantin Burlachenko, Zhize Li, Peter Richtárik MARINA: Faster Non-Convex Distributed Learning with Compression -- ICML 2021
>
> - Peter Richtárik, Igor Sokolov, and Ilyas Fatkhullin. EF21: A new, simpler, theoretically better, and practically faster error feedback -- NIPS 2021

---

> > ### Comment · Reviewer_oTML · 2021-12-01
> > **Response**
> >
> > The authors' response does not address my comment. I agree that communication complexity matters. However, for example, when $\tau $ is close to one, $q$ and $C_q$ will be too large, which does not show how compression helps in terms of communication complexity.
> >
> > Regarding learning rate, for example, the theoretical results in (Alistarh et al. 2017, Theorem 3.5) are obtained under standard learning rates. Of course, the number of iterations is affected by the compression method.

---

> > > ### Author Response · Authors · 2021-12-01
> > > **Response**
> > >
> > > Thank you for your response!
> > >
> > > 1) We have already written about "how compression helps in terms of communication complexity" in another block of response for Reviewer oTML "Remaining questions" , but we can duplicate this text here again:
> > >
> > > Our method gives a benefit in terms of the information transmitted. Our algorithms have the additional factor $(1/q+1/M)$ in bits (=communication) complexity. **This factor can be smaller than 1 for quite large $q$ and $M$, and it means that in bits complexity our method is $1/(1/q+1/M)$ times better!**
> > >
> > >
> > > 2) We studied the paper (Alistarh et al. 2017) again:
> > > About Theorem 3.4:
> > > The stepsize $\eta = 1/(L + \sqrt{K}/\gamma)$. And $\gamma = \frac{\sqrt{2}R}{\sigma \sqrt{T}}$ with $\sigma = B' = \min(n/s^2; \sqrt{n}/s)B$. $s$ is a compression parameter!
> > >
> > > For simplicity, let the step be even larger than in Theorem 3.4 and is equal to
> > > $$\eta = \frac{\gamma}{\sqrt{K}} = \frac{\sqrt{2}R}{\sigma \sqrt{T}\sqrt{K}}.$$
> > > If $s$ becomes smaller (strong compression of information), then $\sigma$ will become larger and $\eta$ will become smaller and smaller! There is a dependence on compression!
> > >
> > >
> > > About Theorem 3.5:
> > > It is very hard to understand what is real stepwise they use. It is written that $\eta = O(1/L)$. But using $O$ notation one can hide the constants responsible for compression. But this is not the main problem of the Theorem. In Theorem 3.5, there is no guarantee of convergence of the method in general. There is a term $\frac{\min(n/s^2; \sqrt{n}/s)B}{L}$, which does not converge with increasing number of iterations $N$.
> > >
> > > For example, in the work
> > >
> > > Eduard Gorbunov, Konstantin Burlachenko, Zhize Li, Peter Richtárik MARINA: Faster Non-Convex Distributed Learning with Compression -- ICML 2021
> > >
> > > where the stepsize depends on the degree of compression, the method converges! What is more important  the big step and no convergence or the theoretically correct step and the presence of convergence?
> > >
> > > Please take into account our comment! We are also waiting for your other questions, because we tried to solve all misunderstandings and problems!

---

> > > > ### Comment · Reviewer_oTML · 2021-12-01
> > > > **Reply**
> > > >
> > > > The learning rate in Theorem 3.4 of  (Alistarh et al. 2017) depends on $\sigma$ whose upper bound depends on $s$, which is an integer and can be controlled ($\sigma$ is bounded).

---

> > > > > ### Author Response · Authors · 2021-12-01
> > > > > **What is the point of this comment?**
> > > > >
> > > > > Dear Reviewer,
> > > > >
> > > > > - We fail to see the point of this comment. How is this relevant to our work? Talking about QSGD, which is a now a famous optimization method, albeit with a suboptimal  (since it's old and many works improve on it), and in the case of Theorem 3.5 incorrect, analysis **is not the most efficient way to talk about the contributions of our paper, which is not even an optimization paper (variational inequalities are more general than optimization).**
> > > > >
> > > > > - Re your comment: Almost all compression mechanisms, including quantization and sparsification, include a parameter which an be used to tune the compression level. You point out that $s$ is the parameter that does this for quantization introduced in Alistarh et al (2017). For Rand-$K$ and Top-$K$ sparsifiers, $K$ is the parameter that does this. The natural dithering quantizer introduced by Horvath et al (2019) [see  arXiv:1905.10988] also has such a parameter, and offer an *exponential* improvement (in terms of variance, if number of levels $s$ is fixed, or in terms of bits in case the variance is fixed) on the quantizer proposed by Alistarh et al. Our analysis a general family of unbiased compression operators which includes the quantization operator from the Alistarh paper, but also the exponentially better quantizer from the Horvath paper. As an example, consider the Rand-$K$ sparsifier. Here we control the compression ratio by choosing $K$ from the values $1,2, \dots, d$. If we follow your "argument" (the point of which we fail to understand as you do not men it clear), we get $q$, $C_q$ and $\tau$ bounded (just like $\sigma$ in your comment), and then the stepsize is bounded. What't the difference of this from what Alistarh et al do? We obtain the same effect. Having said that, this effect is irrelevant. We do not understand why we are even discussing this point.
> > > > >
> > > > > - **Please can you comment on our paper, our rebuttal, and the results we obtained** **We showed to you that we obtain improvement in communication complexity. This what matters. Can you confirm we do, or do you question this? If you agree, then we obtain new theoretical state of the art in communication efficient distributed methods for solving a class of variational inequalities. This is our main contribution.**

---

> > > ### Author Response · Authors · 2021-12-01
> > > **Response to "Response"**
> > >
> > > There is nothing strange going on here. There is no issue with our paper here. Let us explain why the stepsize seems to be small (depending on $q$), and why this is not a problem.
> > >
> > > Let us consider toy example with a $d=4$ dimensional vector.
> > >
> > > - When we apply the $Rand1$ compression mechanism, (we choose a single coordinate uniformly at random) for a 4 dimensional vector, we need to **renormalize** the vector to force the operator to be unbiased (i.e., equal to the original vector in expectation). In this case, if we keep just $k=1$ coordinate out of $d=4$ coordinates, we need to multiply the new vector by $4$. For example, with probability $1/4$ we have $Rand1[(1,2,3,4)]) = 4 \cdot (0,2,0,0)$. So, we **enlarge** the sparsified gradient. **This enlargement needs to be treated by a proper stepsize choice to make the method convergent. This is necessary and not an issue with our method or analysis.**
> > >
> > > - Notice now what can happen. Consider compressing the gradient $g = (1,0,0,0)$. After applying the $Rand1$ compression, we get
> > > $Rand1[(1,0,0,0)]) = 4 \cdot (1,0,0,0)$. **This means that after compression we get a vector that is $4$ (in general $q$) times bigger than the original gradient $g$.** We can't use a GD-like steps here as the method would become divergent, which is why we need a smaller stepsize than $1/L$.
> > >
> > > **About the results from Alistarh et al (2017):**
> > >
> > > - In Theorem 3.4, their learning rate is not $1/L$. It is $1/(L+\sqrt{K}/\gamma)$, where $K$ is the number of parallel workers, and $\gamma$ is a quantity that could be large or small, depending in circumstances. Note that this stepsize is also smaller than the $1/L$ stepsize of GD. It has to be.
> > >
> > > -  Alistarh et al (2017) do not prove Theorem 4.5. They merely state that their result follows easily from the results of Ghadimi et al. However, while Ghadimi et al need the variance to be (uniformly) bounded to obtain their results, quantization does not lead to such a uniformly bounded variance! This is clear from their Lemma 3.1, part (ii). Indeed, quantization leads to a variance bound, but this bound depends on the squared norm of the vector that is being quantized. Gradients and stochastic gradients are not necessarily bounded (consider quadratic functions, for example), and they are not assumed to be bounded in the work of Alistarh et al (2017). **So, their result is, unfortunately, incorrect.** An insight into this is provided in Khaled and Richtarik (2020): https://arxiv.org/abs/2002.03329 . Notice their Section 4.6, where they establish a bound on the 2nd moment of the stochastic gradient obtained by applying a randomized and unbiased compression operator (which included quantizers). The bound is significantly more involved than a constant bound. One can't simply just assume a constant bound - one needs to prove it. Khaled and Richtarik establish convergence in this setting (see their Theorem 2 and Corollary 1). Notice that the stepsize needs to be smaller than $1/L$.  Having said this, nice that Theorem 3.5 in Alistarh et al (2017) is in the nonconvex setting. We do not work with nonconvex problems in our work.

---

> ### Author Response · Authors · 2021-11-23
> **Question 2: discussion following Cor 1**
>
> **Question 2:**
>
> > I think the discussion after Corollary 1 is not accurate. In particular, the condition under which Algorithm 1 outperforms the standard uncompressed extragradient method is restrictive. It is more likely that the standard uncompressed extragradient method outperforms Algorithm 1 in terms of communication complexity.
>
> **Question 2:**
>
> True! There was a simple typo in that part of the paper. We fixed it. Please read the new discussion and our response to Question 1 of Reviewer RfLr where we address the same question.

---

> ### Author Response · Authors · 2021-11-23
> **Question 3:  examples of interesting VI problems beyond saddle point problems**
>
> **Question 3:**
>
> > It will be nice if the authors provide concrete examples of interesting VI problems beyond saddle point problems in machine learning. It is also important to show that Assumptions 1 and 2 hold for such problems. Indeed, the experiments are also based on saddle point problems.
>
> **Response 3:**
>
> - Saddle problems and variational inequalities are a well-known and wide class of problems that have been studied for a long time, and we are clearly not the first to start doing this. We can refer, for example, to the following good book about VIs:
>
> F. Facchinei and J. Pang, Finite-Dimensional Variational Inequalities and Complementarity Problems
>
> We will expand on the discussion in the appendix.
>
>
> - It is also not clear to us why we should specifically check classical saddle point problems and their applications for the fulfillment of Assumptions 1 and 2. Most of the practical examples have already been met in other articles from top conferences:
>
> - Gauthier Gidel, Hugo Berard, Gaëtan Vignoud, Pascal Vincent, Simon Lacoste-Julien, A Variational Inequality Perspective on Generative Adversarial Networks - ICLR 2019
>
> - P Balamurugan, Francis Bach, Stochastic Variance Reduction Methods for Saddle-Point Problems - NIPS 2016
> Yuyang Deng and Mehrdad Mahdavi. Local stochastic gradient descent ascent: Convergence analysis and communication efficiency - AISTATS 2021
>
> - Darina Dvinskikh and Daniil Tiapkin. Improved complexity bounds in wasserstein barycenter problem - AISTATS 2021
> Anatoli Juditsky, Arkadii S. Nemirovskii, and Claire Tauvel. Solving variational inequalities with stochastic mirror-prox algorithm
>
>
> We will mention these in the camera ready version!

---

> ### Author Response · Authors · 2021-11-23
> **Remaining questions**
>
> > “It is unclear how all nodes have access to $F(w^{k+1})$  at the beginning of the iteration $k+1$  while only some nodes with $b_k=1$ compute $F_m(w^{k+1})$?”
>
> - Please look at the end of Algorithm 1:
> Compute $F_m(w^{k+1})$ \& send it to the server; and get $F(w^{k+1})$ from the server. We assume the server can broadcast at low communication complexity (a common assumption in virtually all papers in distributed optimization with compression), or that the server does not exist physically, and is merely an abstraction for "all other nodes", in which case the server does not need to communicate back to the nodes.
>
> - Devices send to the server their $F_m(w^{k+1})$,  and the server averages these vectors, computes  $F(w^{k+1})$, and broadcasts this back to the devices. This process happens without compression, but we need it only a few times, because $\tau$ is close to 1 (see also the comment above).  The same trick with full gradient is used in paper about minimization
>
> Eduard Gorbunov, Konstantin Burlachenko, Zhize Li, Peter Richtárik MARINA: Faster Non-Convex Distributed Learning with Compression -- ICML 2021
>
> > In Table 1, the authors mentioned that ``(Deng & Mahdavi 2021) does not have strong theory since it cannot improve on non-compressed methods without assuming data homogeneity.'' I my view, a similar criticism applies to this work too.
>
> - The main problem of Deng & Mahdavi 2021 is that they use Descent-Ascent without Extra Step and this method diverges for simple bilinear saddle point problems in theory and in practice (see our experiments). We change Table 1 appropriately to make sure this misunderstanding is avoided.
>
> - About homogeneity: if it is about the previous point of how all nodes have access to $F(w^{k+1})$, we explained it in the previous paragraph.
>
> - About “cannot improve on non-compressed methods”: See the new version of discussion after Corollary 1. Our algorithms have the factor $(1/q+1/M)$ in bits (=communication) complexity. **This factor can be smaller than 1 for quite large $q$ and $M$, and it means that in bits complexity our method is $1/(1/q+1/M)$ times better!**
>
> > "we introduce the notion of expected density, defined ..." This is not new.
>
> - We know that is not new. Sorry, for misunderstanding, we will fix this wording. We think “define” or “denote” are ok.
>
> >Typos and minors
>
> We fixed this in the new version of the paper.
>
> To sum up, all the weaknesses regarding our paper are either misunderstandings or small problems that are easy to fix in the final version. We do not understand why our article was rejected and we hope that the rating will increase.

---

### Official Review · Reviewer_n8kw · 2021-11-02

**Correctness:** 3
**Technical Novelty And Significance:** 2
**Empirical Novelty And Significance:** 2
**Recommendation:** 5
**Confidence:** 2

**Main Review:**

This paper proposes compression methods for solving variational inequalities. It is good to cover various settings such as unbiased vs. contractive, deterministic vs. stochastic. Detailed convergence analysis is also provided. However, I also have the following concerns.

1. For practical problems such as GAN or adversarial training, the objective of min-max optimization is nonconvex-nonconcave, which results in a non-monotonic operator in variational inequalities. This more practical assumption has been considered in existing work with sound theoretical analysis (see reference [1] as below). The authors are encouraged to extend the analysis to the more practical nonconvex-nonconcave regime.

2. In the paragraph right above Figure 2, the authors mentioned that “the learning curves on Figure 1(upper) follow a predictable pattern, with more extreme compression techniques demonstrating slower per-iteration convergence”. First, is Figure 1(upper) a typo? Should it be Figure 2(upper). Second, the learning curves in Figure 2(upper) are overlapping with each other (except for pure 8-bit in the upper-left figure). I am not sure how the authors get the observation that  “more extreme compression techniques demonstrating slower per-iteration convergence”. Besides, some discussion is needed to compare the compression ratio between 8-bit quantization and power compression with rank r=8. This is because power compression with rank r would decompose the dense matrix of size m\*n into two matrices of size
m\*r and n\*r. The exact compression ratio would depend on the values of m and n.

3. In Figure 2(lower), it’s customary to also list average scores across different tasks in GLUE benchmark. It’s easier to compare different methods by checking their corresponding average scores.

4. In Figure 1, it looks like that extra-step method also diverges (i.e., accuracy decreases as training goes). Since the extra-step method is specific for solving saddle point problems, it’s counter-intuitive when it diverges.

5. It looks like there is no conclusion section at the end of this paper.

Reference:
[1] Diakonikolas, Jelena, Constantinos Daskalakis, and Michael Jordan. "Efficient methods for structured nonconvex-nonconcave min-max optimization." International Conference on Artificial Intelligence and Statistics. PMLR, 2021.



**Summary Of The Paper:**

This paper considers the compression methods for solving variational inequalities (or saddle point problems in particular) in the distributed setting. Both unbiased (i.e., MASHA1) and contractive (i.e., MASHA2) compression methods have been proposed. Theoretical analysis is provided to show that the proposed method can converge.


**Summary Of The Review:**

I have been working on related areas and have read this paper carefully.

---

> ### Author Response · Authors · 2021-11-23
> **Question 1: Nonconvex setup**
>
> **Question 1:**
>
> > For practical problems such as GAN or adversarial training, the objective of min-max optimization is nonconvex-nonconcave, which results in a non-monotonic operator in variational inequalities. This more practical assumption has been considered in existing work with sound theoretical analysis (see reference [1] as below). The authors are encouraged to extend the analysis to the more practical nonconvex-nonconcave regime.
>
> **Response 1:**
>
> - **This is an interesting question for future work. Indeed, the theory of methods for non-monotone inequalities is poorly developed now. In particular, you cite one of the leading works in this area*. If we look at this paper, the authors of this work position it as purely theoretical and do not even try to do experiments.
>
>
> - Moreover, **in the community about saddle point problems, it is normal to make a theory in the convex case and use it for GANs:**
>
> Gauthier Gidel, Hugo Berard, Gaëtan Vignoud, Pascal Vincent, Simon Lacoste-Julien, A Variational Inequality Perspective on Generative Adversarial Network; ICLR 2019
>
> - Finally, in our opinion, **while the nonconvex-nonconcave case is more general,  in fact it typically gives very weak convergence guarantees which do not correspond to what happens in practice.** For practical neural networks, the situations is  much more favorable from the point of view of empirical convergence, both from the point of view of speed and from the point of view of the solution quality.  So, there is something else at play, and it may be the case that training often behaves as if the problem was convex-concave. So, the theory we provide is not irrelevant to practice.
>
>
> In summary, while we believe that **studying the nonconvex-nonconcave case  is an interesting direction for future work, it is simply not the focus of the current paper,  and should not be used as a reason to reject it.** Progress happens gradually, and simpler problems need to be understood before we can attack harder problems.  **We wish to suggest that our paper is judged by the results it contains, and not the results it does not contain.** Each paper needs to have a focus, and no paper will consider all possible scenarios, generalizations and extensions.

---

> ### Author Response · Authors · 2021-11-23
> **Question 2a: paragraph above Fig 2**
>
> **Question 2a:**
>
> > In the paragraph right above Figure 2, the authors mentioned that “the learning curves on Figure 1(upper) follow a predictable pattern, with more extreme compression techniques demonstrating slower per-iteration convergence”. First, is Figure 1(upper) a typo? Should it be Figure 2(upper).
>
> **Response 2a:**
>
> Thanks, this is a typo - Figure 2 (upper).  We fixed it.

---

> ### Author Response · Authors · 2021-11-23
> **Question 2b: learning curves**
>
> **Question 2b:**
>
> > Second, the learning curves in Figure 2(upper) are overlapping with each other (except for pure 8-bit in the upper-left figure). I am not sure how the authors get the observation that “more extreme compression techniques demonstrating slower per-iteration convergence”.
>
> **Response 2b:**
>
> While there is some overlap due to the inner stochasticity of the methods, the average objective for all compression methods was in fact consistent. You can see it in Figure 2. To better illustrate that, we changed Figure 2 (upper right) in the new version of the paper, and zoomed it more. So, all is as one would expect.

---

> ### Author Response · Authors · 2021-11-23
> **Question 2c: 8-bit quantization and power compression**
>
> **Question 2c:**
>
> > Besides, some discussion is needed to compare the compression ratio between 8-bit quantization and power compression with rank $r=8$.
>
> **Response 2c:**
>
> We concur and provide the discussion as requested (we will add it to the paper).
>
> For this evaluation, we followed the compression strategy from the original PowerSGD paper. Namely, we apply compression to the weight matrices, but keep all 1d parameters intact. These 1d parameters consist of biases and LayerNorm scales which constitute less than 1\% of all model parameters. The resulting procedure reduces the total amount of communication by a factor of $12.32\times$. In practice, we observed that the communication time is only $9.1\pm 0.74$ times faster due the computation overhead.

---

> ### Author Response · Authors · 2021-11-23
> **Question 3: average scores across different tasks in GLUE benchmark**
>
> **Question 3:**
>
> > In Figure 2(lower), it’s customary to also list average scores across different tasks in GLUE benchmark. It’s easier to compare different methods by checking their corresponding average scores.
>
> **Response 3:**
>
> Of course; we have added it to the new version of the paper. Please, look. Thanks!

---

> ### Author Response · Authors · 2021-11-23
> **Question 4: Fig 1 - extra-step method diverges?**
>
> **Question 4:**
>
> > In Figure 1, it looks like that extra-step method also diverges (i.e., accuracy decreases as training goes). Since the extra-step method is specific for solving saddle point problems, it’s counter-intuitive when it diverges.
>
> **Response 4:**
>
> This is a misunderstanding. Only the method which is based on SGD diverges! **The Extra Step method with quantization (violet)  converges, but it is slower than our new methods.**

---

> ### Author Response · Authors · 2021-11-23
> **Question 5: Missing conclusion**
>
> **Question 5:**
>
> > It looks like there is no conclusion section at the end of this paper.
>
> **Response 5:**
>
> - Conclusion is not a mandatory section. However, we are happy to add it to the camera ready version of the paper. Here is the text:
>
> "We presented the first distributed methods with compression for variational inequalities and saddle point problems. The methods support the use of both unbiased and biased compressors. Proofs of convergence in the convex-concave and strongly-convex-strongly-concave cases were given. Meanwhile, our work points to several possibilities for future work. For example, it is interesting to study the non-convex case. However, this case is very challenging even in the non-distributed setup. The other issue of extending these methods from a centralized network to a decentralized topology setup is also interesting."
>
>
> ---
>
> In summary, **while all critical points raised by the reviewer are valid, they are very very minor. We do not understand why our work is rated so low.**

---

### Official Review · Reviewer_V1y7 · 2021-11-03

**Correctness:** 2
**Technical Novelty And Significance:** 2
**Empirical Novelty And Significance:** 2
**Recommendation:** 3
**Confidence:** 5

**Main Review:**

Main techniques used and the good points about the paper
Both the algorithms basically extend the extragradient/extrastep method of solving variational inequalities to the distributed setting. So in particular they specify how this method can be modified to fit in a distributed regime and prove the convergence and give upper bounds on the total communication.

Main drawbacks and technical issues with the paper
I think this is an interesting modification of the prior known work but I do not believe that this is a significant research contribution and is an incremental adaptation of the extragradient/extrastep method. Also the convergence or upper bounds on communication while cumbersome are not non trivial and straightforward adaptations of the classical centralized setting proofs. I think a worthwhile theoretical contribution would be to prove lower bounds on how much communication overhead would be needed to solve variational inequalities in the distributed setting.  Or at the bare minimum have some kind of restriction assumptions on the algorithm and then prove lower bounds that could explain why this adaptation of the extragradient method is near optimal or non trivial.

Presentation of the paper.
I think it would help the reader to first understand the classical solution to the variational inequality problem before jumping into the distributed solution. Section 4 is really abrupt and it would help to ease the reader into the distributed solution. Otherwise the paper looks okay.

**Summary Of The Paper:**

The paper studies the communication needed in order for a group of distributed players to collectively solve a variational inequality problem. The paper provides two algorithms MASHA1, MASHA2 to do this that solve both the deterministic and stochastic cases.  They also provide experimental results on applying their techniques to Bilinear saddle point problem and adversarial training of transformers.



**Summary Of The Review:**

The paper studies the communication needed in order for a group of distributed players to collectively solve a variational inequality problem. In particular they provide two algorithms MASHA1, MASHA2 to do this that solve both the deterministic and stochastic cases.
They also provide experimental results on applying their techniques to Bilinear saddle point problem and adversarial training of transformers.
Both the algorithms basically extend the extragradient/extrastep method of solving variational inequalities to the distributed setting. I think this is an interesting modification of the prior known work but I do not believe that this is a significant research contribution and is an incremental adaptation of the extragradient/extrastep method.

---

> ### Author Response · Authors · 2021-11-23
> **Question 1: Incremental research**
>
> **Question 1:**
>
> > Main drawbacks and technical issues with the paper I think this is an interesting modification of the prior known work but I do not believe that this is a significant research contribution and is an incremental adaptation of the extragradient/extrastep method. Also the convergence or upper bounds on communication while cumbersome are not non trivial and straightforward adaptations of the classical centralized setting proofs. I think a worthwhile theoretical contribution would be to prove lower bounds on how much communication overhead would be needed to solve variational inequalities in the distributed setting. Or at the bare minimum have some kind of restriction assumptions on the algorithm and then prove lower bounds that could explain why this adaptation of the extragradient method is near optimal or non trivial.
>
> > I think this is an interesting modification of the prior known work but I do not believe that this is a significant research contribution and is an incremental adaptation of the extragradient/extrastep method.
>
> **Response 1:**
>
> 1) We first wish to point out that this reviewer merely **claims** that our method is a simple/straightforward adaptation of existing methods and uses this claim to suggest our work is incremental research, and that incremental research should not be accepted to ICLR. **No justification is provided whatsoever.** **We wish to flag this to the AC.** It should not be allowed for reviewers to claim that some work is a simple adaptation of existing works without providing evidence. Otherwise every single paper can be rejected using the exact same logic. All research is incremental in the correct sense of the word, the question is how large, non-trivial, important and impactful a step the paper makes. Some examples: ADAM is a simple modification of SGD with momentum, Nesterov gradient descent is a simple modification of gradient descent, GANs are really just min-max problems studied before, and so on. **We ask other reviewers and the AC to come to our defense.**
>
> 2) We emphasize that saddle point problems are very popular in the literature. Moreover, virtually all contribution around variational inequalities is “incremental adaptation” of the extragradient method (moreover, these works are quite popular in the community and have been presented in well-known venues). Let us take a look at these examples:
>
> - The **original extragradient method** was proposed in G. M. Korpelevich in "The extragradient method for finding saddle points and other problems".
> $$z^{k+1/2} = {\rm pr}[z^k - \gamma F(z^k)]$$
> $$z^{k+1} = {\rm pr}[z^k - \gamma F(z^{k+1/2})]$$
>
> - In their influential paper "Solving variational inequalities with stochastic
> mirror-prox algorithm", Anatoli Juditsky, Arkadii S. Nemirovskii, and Claire Tauvel   presented a **stochastic version of extragradient in any geometric setup**:
> $$z^{k+1/2} = {\rm pr}[z^k - \gamma F(z^k, \xi^k)]$$
> $$z^{k+1} = {\rm pr}[z^k - \gamma F(z^{k+1/2}, \xi^{k+1/2})]$$
>
> This can be considered a small/incremental research. Of course, that would be a meaningless evaluation of this great paper.
>
> - In the  ICLR 2019 paper "A Variational Inequality Perspective on Generative Adversarial Networks", Gauthier Gidel, Hugo Berard, Gaëtan Vignoud, Pascal Vincent, and Simon Lacoste-Julien presented the **Past Extra Gradient Method**:
> $$z^{k+1/2} = {\rm pr}[z^k - \gamma F(z^{k-1/2})]$$
> $$z^{k+1} = {\rm pr}[z^k - \gamma F(z^{k+1/2})]$$
>
> Notice how small the change is. Small changes often lead to massive improvements, difficult challenges, and progress.
>
> - In the NeurIPS 2019 paper "On the Convergence of Single-Call Stochastic Extra-Gradient Methods" by Yu-Guan Hsieh, Franck Iutzeler, Jérôme Malick, Panayotis Mertikopoulos, the authors
> proposed the **Reflected Extra Gradient Method**:
> $$z^{k+1/2} = 2z^k - z^{k-1}$$
> $$z^{k+1} = {\rm proj}[z^k - \gamma F(z^{k+1/2})]$$
>
> Again, the changes may seem small to a non-expert. Yet, this is a significant contribution to the field.
>
> 3) The above are just a few examples of well-known papers about VIs. All these articles can be labeled as “incremental adaptation” of the extragradient method if one wants to reject them. However, that is not how academic/scientific reviewing should work. The community appreciates these works, and they all lead to important advances in the field. We hope that now the Reviewer will appreciate our contribution to the development of VIs. **We are the first to turn to VI methods with compression, and obtained a very interesting, in our view, modification of the extragradient method, with theoretical communication complexity benefits.** **Our methods are current SOTA for solving VIs in distributed regime in terms of theoretical communication complexity.** Do we want SOTA methods to be labeled incremental and reject them?

---

> ### Author Response · Authors · 2021-11-23
> **Question 2: Lower bounds**
>
> **Question 2:**
>
> > I think a worthwhile theoretical contribution would be to prove lower bounds on how much communication overhead would be needed to solve variational inequalities in the distributed setting. Or at the bare minimum have some kind of restriction assumptions on the algorithm and then prove lower bounds that could explain why this adaptation of the extragradient method is near optimal or non trivial.
>
> **Response 2:**
>
> - **This is interesting question, but please note that there are no lower bounds even for minimization problems with quantization! This is an important open problem on its own, but not the problem we address in this paper. We would want to request that our paper be judged by the results we actually obtain.**
>
> - Very few papers on optimization or VIs prove lower bounds. Does this mean that they should be rejected? No. We do not claim that you believe this, but we want to be clear about our opinion wrt this criticism. This is not a criticism which should in any way affect the score of our paper. We take it as an interesting suggestion for future research. As we explain above, the problem you want us to study is difficult and was not resolved even in the simpler setting of optimization.

---

> ### Author Response · Authors · 2021-11-23
> **Question 3: Presentation of the paper**
>
> **Question 3:**
>
> > Presentation of the paper. I think it would help the reader to first understand the classical solution to the variational inequality problem before jumping into the distributed solution. Section 4 is really abrupt and it would help to ease the reader into the distributed solution. Otherwise the paper looks okay.
>
>
> **Response 3:**
>
> - VIs are a fairly popular class of problems. Many articles have been written about non-distributed methods for VIs. One can read, for example, those that we cite in the paper or mention in a previous comment.
>
> - **Our article is about distributed methods, we would simply  not be able to  meet the page limit if we were to write a detailed overview of non-distributed methods. We had to make some choices here, and opted to focus on what is novel here rather than on introducing VIs as such. The main part of the paper is intended for experts in saddle point problems and in compression techniques. Therefore, a good background is assumed.**
>
> - However, we of course agree that if there was no page limit, inclusion of some commentary on the classical approach would be very useful. **We shall include such an introduction in the appendix!**
>
> **In summary, this is a very minor comment which is unrelated to the scientific technical contribution of our paper. As such, we view it as a small and useful suggestion, but not as a piece of criticism which should in any effect be reflected in the score of our pawer, let alone on the accept/reject decision.**

---

### Official Review · Reviewer_RfLr · 2021-11-08

**Correctness:** 2
**Technical Novelty And Significance:** 3
**Empirical Novelty And Significance:** 3
**Recommendation:** 5
**Confidence:** 3

**Main Review:**

The authors study a well motivated problem. The algorithm developed has a solid theoretical framework from an optimization theory viewpoint. The convergence results seem non-trivial and thorough. However, some concerns exist.

- A critical question that remains unanswered - from the viewpoint of theory - is whether there exist compressors that indeed lead to improved communication complexity over the uncompressed methods. In particular, the authors report that it is needed that $\beta q^{(1+q/M)} < 1$ is required. However, for the randK compressor where $\beta = (M+1)/Mq$ - assuming that the same compressor everywhere -  this relation is not satisfied. To complete the arguments purported in the paper, it seems necessary to provide this analysis for the randK compressor, or some other compressor that performs well.

- The experimental section appears somewhat weak, I would have liked to see more.. Although the paper is motivated via GANs and adversarial training, results are only provided for the latter. Perhaps more importantly, the influence of various parameters (compressors, compression ratio, tau) on the performance is not studied. In effect, the experiments do not appear to be a sufficiently thorough validation of the theory.

- The presentation can be improved by presenting and explaining the idea behind the basic centralized solution in the appendix, or before presenting the solution with the decentralized, compressed version.

- From the communication complextity reported below Corollary 1, it is surprising that the compression-factor of the devices influences the total number of bits much more than the compression factor at the  server, essentially the communication complexity appears to be 2^{(1+q1)*log(q2)} bits, where q1,q2 respectively represent the variances of the quantizers of the devices and the servers respectively. It is also not transparent that this disparity in behaviour exists looking at Theorem 1, where the dependence on q1,q2 appears more symmetric (at least in an order sense). Please explain.



**Summary Of The Paper:**

The paper develops a decentralized algorithm for solving variational inequalities with a certain structure motivated by machine learning applications. The key innovation is the utilization of compression (i.e., quantization) for communicating loss functions and their aggregates between a set of devices and a centralized server node, towards solving the inequalities using extragradient methods. The paper provides theoretical convergence guarantees that account for the quantization errors, and report the trade-off between the communication complexity    and the convergence.

**Summary Of The Review:**

Overall, non-trivial theoretical analysis has been conducted to study the effect of compression on convergence. However, some concerns exist in terms of both theoretical justification need to be addressed. The experimental analysis could have been more thorough, though I consider this a relatively minor weakness given the extent of theoretical contribution in terms of convergence analyses.

---

> ### Author Response · Authors · 2021-11-23
> **Question 1: improved communication complexity**
>
> **Question 1:**
>
> > A critical question that remains unanswered - from the viewpoint of theory - is whether there exist compressors that indeed lead to improved communication complexity over the uncompressed methods. In particular, the authors report that it is needed that $\beta q (1+q/M)<1$  is required. However, for the randK compressor where $\beta = (M+1)/Mq $  - assuming that the same compressor everywhere - this relation is not satisfied. To complete the arguments purported in the paper, it seems necessary to provide this analysis for the randK compressor, or some other compressor that performs well.
>
> **Response 1:**
>
> - Yes, of course **there exist such compressors - we do not merely get an improvement in practice, but also in theory.** We have now added an explanation in a new paragraph following Corollary 1. The new paragraph is colored in orange.
>
> - There was a **typo** in our paper in the expression you refer to; this is perhaps what led to the confusion. $q^{\text{serv} (1 + q/M)}$   should have been $q^{\text{serv}} (1 + q/M)$ (we misplaces one curly bracket). We apologize. We fixed this. We wish to point out that this typo was isolated - it appeared in one place only. Everywhere else where this expression was men toned, e.g., Table 1, the expression was correct.
>
> **In summary, we believe this issue is not critical at all; and was based on an isolated typo due to one misplaces curly bracket. Plus we did not explicitly mention theoretical improvement using compression, which of course we should have done, and we have fixed this now. We believe this issue  is now fully resolved.** If any further issues related to this concern remain, we are most happy to respond and clarify. Please ask.

---

> ### Author Response · Authors · 2021-11-23
> **Question 2: experiments**
>
> **Question 2:**
>
> > The experimental section appears somewhat weak, I would have liked to see more.. Although the paper is motivated via GANs and adversarial training, results are only provided for the latter. Perhaps more importantly, the influence of various parameters (compressors, compression ratio, tau) on the performance is not studied. In effect, the experiments do not appear to be a sufficiently thorough validation of the theory.
>
> **Response 2:**
>
> Thanks for this comment.
>
> 1) **We are of the belief that theoretical papers should be primarily evaluated from the point of view of theory, not experiments.** Conversely, empirical papers should be primarily evaluated based on the empirical results, and not judged harshly if theory is missing. This is just a basic principle of symmetry highlighting the importance of both theoretical and empirical research. We would therefore appreciate if you could re-evaluate the impact of this criticism on our score.
>
> 2) **Our paper is primarily theoretical, and the experiments are mainly serving the purpose of illustrating these results. Having said that, we believe that our experimental results are of an unusually high standard for a theoretical paper.** Indeed, we have also performed a real (i.e., not simulated) experiment to train a large model. ALBERT is really worth training in a distributed manner. We wish to most articles about the theory of methods with compression are limited to simpler simulations on one machine, here are some examples:
>
> - *NeurIPS 2021:* "EF21: A new, simpler, theoretically better, and practically faster error feedback":  ResNet + CIFAR10
>
> - *ICML 2021:* "MARINA: Faster Non-Convex Distributed Learning with Compression": small two-layer neural networks + LIBSVM datasets
>
> - *NeurIPS 2020:* "Linearly converging error compensated SGD": logistic regression + LIBSVM datasets
>
>
> 3) Re "the influence of various parameters (compressors, compression ratio, tau) on the performance is not studied": **We agree this is a good idea. We are currently running such experiments.** We will not able to complete them by the deadline, but we will certainly include them in the camera-ready version of the paper should the paper be accepted.
>
> In summary, we argue that this comment should be seen as minor, and in our view it should not have an influence on the acceptance decision.

---

> ### Author Response · Authors · 2021-11-23
> **Question 3: presentation**
>
> **Question 3:**
>
> > The presentation can be improved by presenting and explaining the idea behind the basic centralized solution in the appendix, or before presenting the solution with the decentralized, compressed version.
>
> **Response 3:**
>
> - This is a good idea, but we believe it is a very minor concern. We already modified some parts of the paper, and will work on the appendix as well.
>
>
>
> - Please note though that due to the severe page restrictions, we had to make hard some hard choices on what to include. We opted to include more original results are include less introductory material as our paper is aimed at experts in the field who are expected to be familiar with these topics. Having said that, we will elaborate on the examples section (Section D) and expand it to serve as a brief introduction to the field.
>
> - We believe the explanation in Section 4.1 is sufficient.

---

> ### Author Response · Authors · 2021-11-23
> **Question 4: communication complexity reported below Corollary 1**
>
> **Question 4:**
>
> > From the communication complexity reported below Corollary 1, it is surprising that the compression-factor of the devices influences the total number of bits much more than the compression factor at the server, essentially the communication complexity appears to be $2^{(1+q_1)*log(q_2)}$ bits, where $q_1, q_2$ respectively represent the variances of the quantizers of the devices and the servers respectively. It is also not transparent that this disparity in behaviour exists looking at Theorem 1, where the dependence on $q_1, q_2$ appears more symmetric (at least in an order sense). Please explain.
>
> **Response 4:**
>
> We clarified this point in our response to Question 1, and we also changed the text following Corollary 1 in the paper. There was a small typo in one of the expressions, all is OK after this type is fixed.

---

### Author Response · Authors · 2021-11-23
**New version of the paper**

We uploaded a new version of the paper. We tried to take into account all the comments of Reviewers. Major changes are highlighted in orange.

---

### Decision · Program_Chairs · 2022-01-20

**Decision:**

Reject

**Comment:**

In this paper, the authors consider two algorithms for solving (strongly) monotone variational inequalities with compressed communication guarantees, MASHA1 and MASHA2. MASHA1 is a variant of a recent algorithm proposed by Alacaoglu and Malitsky, while MASHA2 is a variant of MASHA1 that relies on contractive compressors (by contrast, MASHA1 only involves unbiased compressors). The authors then show that
- MASHA1 converges at a linear rate (in terms of distance to a solution squared), and at a $1/k$ rate when taking its ergodic averge (in terms of the standard VI gap function).
- MASHA2 converges at a linear rate (in terms of distance to a solution squared).

Even though the paper's premise is interesting, the reviewers raised several concerns which were only partially addressed by the authors' rebuttal. One such concern is that the improvement over existing methods is a multiplicative factor of the order of $\mathcal{O}(\sqrt{1/q + 1/M})$ in terms of communication complexity (number of transmitted bits) for the RandK compressor, which was not deemed sufficiently substantive in a VI setting (relative to e.g., wall-clock time, which is not discussed).

After the discussion with the reviewers during the rebuttal phase, the paper was not championed and it was decided to make a borderline "reject" recommendation. At the same time, I would strongly urge the authors to resubmit a properly revised version of their paper at the next opportunity (describing in more detail the innovations from the template method of Alacaoglu and Malitsky, as well as including a more comprehensive cost-benefit discussion of the stated improvements for the RandK/TopK compressors).